# Cervical cancer: Riverside women's knowledge in the Brazilian Amazon about preventive measures

Ricardo Luiz Saldanha da Silva[iD][1]*, Shirley Regina Cardoso Mendes[2],
Bianca Silva de Brito[3], Bianca Pimentel de Moura[4], Ivaneide Leal Ataíde Rodrigues[1],
Erlon Gabriel Rego de Andrade[1], Eliene do Socorro da Silva Santos[1],
Laura Maria Vidal Nogueira[1]

1 Graduate Program in Nursing, Escola de Enfermagem Magalhães Barata, Universidade do Estado do Pará, Belém, Pará, Brazil, 2 Graduate Course in Nursing, Escola de Enfermagem Magalhães Barata, Universidade do Estado do Pará, Belém, Pará, Brazil, 3 Oncology Nursing Residency, Uniprofessional Health Residency Program, Universidade do Estado do Pará, Belém, Pará, Brazil, 4 Oncology Nursing Residency, Multiprofessional Health Residency Program, A.C. Camargo Cancer Center, São Paulo, Brazil

* ricardos.enf2018@gmail.com

## Abstract

### Objective

To analyze riverside women's knowledge about preventive measures against cervical cancer and its influence on the adoption of these measures.

### Methods

This study used a descriptive, qualitative design, carried out with 20 riverside women at the Municipal Health Unit on the island of Cotijuba, municipality of Belém, state of Pará, Brazil. Data were produced from April to July 2023, through individual interviews with a semi-structured script, consisting of questions to understand participant sociodemographic characteristics and explore their subjectivities regarding the object of study. Descriptive statistical analysis of sociodemographic data was carried out. Subjective data were transcribed to form a text *corpus*, subjected to lexical processing and analysis with *Interface de R pour les Analyzes Multidimensionnelles de Textes et de Questionnaires*, version 0.7, alpha 2.

### Results

Among the participants, ages ranged from 23 to 64 years old and the age group of 40–64 years old prevailed (n = 11; 55%). After using the software, we identified 262 text segments, using 222 (84.73% of the *corpus*), generating six lexical classes, organized into three thematic axes. These axes demonstrate riverside women's perceptions, behaviors and access to information regarding cervical cancer as well as their knowledge about preventive measures, the human papillomavirus and its

**Data availability statement:** All relevant data are publicly available from the Inter-university Consortium for Political and Social Research (ICPSR) at the following DOI: https://doi.org/10.3886/E229961V1.

**Funding:** This work was supported by two scholarships granted through the Institutional Scientific Initiation Scholarship Program of the Universidade do Estado do Pará (Notice 046/2022). The Brazilian National Council for Scientific and Technological Development (*Conselho Nacional de Desenvolvimento Científico e Tecnológico* – CNPq, https://www.gov.br/cnpq/pt-br) supported author RLSS (Process 153917/2022-9), and the Amazon Foundation to Support Studies and Research (*Fundação Amazônia de Amparo a Estudos e Pesquisas* – FAPESPA, https://www.fapespa.pa.gov.br/) supported author SRCM (Registration 0048397). The funders had no role in study design, data collection and analysis, decision to publish, or preparation of the manuscript.

**Competing interests:** The authors have declared that no competing interests exist.

relationship with this type of cancer. Gaps were identified in women's knowledge about the disease, regarding the correct way to protect themselves to avoid cervical cancer.

## Conclusions

Particularities of thought and attitude were observed in the adoption of preventive measures, a context in which it was possible to reflect on the way in which these aspects had or could have an impact on women's daily lives, in order to strengthen or weaken their health.

---

## Introduction

Among malignant neoplasms, cervical cancer (CC) is an important public health concern, being one of the main causes of death among women [1,2]. It is characterized by the abnormal growth of cells, generated especially by persistent infection with oncogenic Human Papillomavirus (HPV) of various groups. These viruses have the ability to specifically infect epithelial cells and replicate in the nucleus of squamous cells, particularly those located in the genital region [3–5].

The prevalence of HPV infection can vary significantly depending on the population studied and the methods used to detect the virus. Studies have shown high rates of this infection among women living in the Amazon region of Brazil [6–12], a context in which a meta-analysis showed an average prevalence of 25.41% for cervical HPV in Brazilian regions, with particularly high rates in populations with less access to health services [6] and in women aged 25 or younger [7–9].

In riverside populations in the state of Pará, one of the federative units of this country, the prevalence of HPV was estimated at 16.4%, with 2.3% of cases attributed to HPV-16 and 1.4% to HPV-18, both with high oncogenic risk, and approximately 70% of women had never had a Pap smear (PS) [7,8]. Low screening coverage, combined with the existence of viral types with high oncogenic risk, indicates increased vulnerability to the development of cervical lesions in these populations [9].

In the context of traditional communities and peoples, epidemiological analyses also highlight a high prevalence of HPV infection among indigenous people in the Amazon region, with an estimated rate of 34.2%. This highlights the need for culturally appropriate public health actions and strategies to reduce the occurrence of this infection and its biopsychosocial repercussions among these communities [9], whose vulnerabilities tend to be as severe as those found in the daily lives of riverside populations.

Among the ways to prevent HPV infection and, consequently, the risks of becoming ill from CC, using condoms and anti-HPV vaccination stand out. Furthermore, the pathology is directly linked to smoking, prolonged use of oral contraceptives, early onset of sexual activity and a sexual lifestyle with multiple partners. Usually, signs and symptoms are not evident in the initial phase of the disease, but, as it worsens,

manifestations such as intermittent vaginal bleeding or bleeding after sexual intercourse, abnormal vaginal discharge and abdominal pain may occur [13,14].

With PS, early diagnosis can be made and possible worsening can be prevented [4,5,15]. It is worth highlighting that, in addition, other strategies can be used to detect HPV early, such as innovative self-collection methods combined with molecular diagnosis, which have been shown to be an effective strategy for screening for CC among women in rural and hard-to-reach communities, such as riverside populations [16–18]. Studies conducted in the Northern region of Brazil, especially in the states of Amazonas and Pará, showed high acceptability of cervicovaginal self-collection among women who participated in these studies, in addition to significant agreement between self-collected samples and samples collected by health professionals to detect HPV and high-risk HPV [16–18].

According to global estimates, this type of cancer ranks fourth in the highest incidence in the female population, presenting more than 604,000 new cases in 2020, which corresponds to 3.1% of all types of cancer, accounting for the seventh most common in the world [19]. In Brazil, from 2023 to 2025, more than 17,000 new cases are expected, with an estimated risk of 15.38 cases for every 100,000 women [20].

Unequally distributed among Brazilian regions, CC has a higher prevalence in the North, Northeast and Central-West due to the peculiarities of these regions. In the North region, according to national estimates for the 2023–2025 period, CC ranks third among the most common types of cancer [20]. It mainly affects women aged 25 and older, and is associated, in most cases, with challenging sociodemographic and epidemiological conditions, such as low income, reduced access to health promotion and illness prevention actions, in addition to a low level of education [21].

In the context of the groups that inhabit the Brazilian territory, riverside populations constitute a segment exposed to social vulnerabilities that are characterized by sociodemographic and epidemiological conditions similar to those already mentioned. This reflection is made considering the assistance initiatives that are insufficiently received from the Brazilian State through its public policies, contributing to reduced or unfeasible access to health services [22,23]. Furthermore, it is necessary to consider the scenario in which riverside women experience certain restrictions in their own healthcare due, for instance, to a low level of knowledge about the importance of certain services or preventive procedures [7].

Therefore, the issues at hand indicate the need to better assist this population, considering that CC represents a concern in women's daily lives, mainly in developing countries and among vulnerable groups, such as riverside populations, as prevention and control measures have not yet achieved encouraging results [24].

Given the relevance of the topic, this study aimed to analyze riverside women's knowledge about preventive measures against CC and its influence on the adoption of these measures.

## Methods

### Study design and setting

We carried out a descriptive and qualitative study, whose writing met the COnsolidated criteria for REporting Qualitative research (COREQ) guidelines [25]. It was carried out at the Municipal Health Unit (MHU) on the island of Cotijuba, located in the Administrative District of Outeiro (DAOUT), one of the eight Administrative Districts that configure the management organization of the municipality of Belém, state of Pará, Brazil [26,27].

This unit is managed by the Municipal Health Department of Belém (In Portuguese, *Secretaria Municipal de Saúde de Belém*, SESMA) [28], being the only health establishment available on the island to assist the local population. It offers basic care services, laboratory tests and maternal and child care, in addition to emergency care. The healthcare team is made up of community health workers (CHW) and approximately six nurses, four doctors, 19 nursing technicians and a nursing assistant.

In this scenario, it is worth highlighting that access to the island of Cotijuba occurs only by river and that local tourism constitutes its main economic activity. Furthermore, there is a lack of public investment to qualify access to actions and

services that affect living and health conditions, such as drinking water supply, solid waste collection, electricity supply and paving of public roads [29].

## Data source

A total of 20 riverside women registered at the MHU on the island of Cotijuba participated. This quantity was defined by data saturation, identified by the researchers when new elements were not found in participants' reports and were no longer necessary to understand the object of study [30].

We included women aged 18–64, who had physical and cognitive conditions to be interviewed. This borderline age group considers the target age to perform PS, in accordance with guidelines from the Brazilian Ministry of Health, which recommends the test for women aged 25 years or older [5]. However, we also included women aged below the age group recommended by the Ministry of Health due to evidence that points to early onset of female sexual life, as demonstrated in an integrative literature review, which addressed the forms of HPV transmission among adolescents. Furthermore, it is understood that this fact is associated with the weakness in protecting women during early onset sexual practices, exposing them to HPV infection and the consequent risk of becoming ill due to CC [31].

It was decided to exclude those who had not yet started their sexual life and those with whom it was not possible to schedule a suitable day and time to carry out the interview, after three attempts during the data production period. However, there were no exclusions, refusals or withdrawals.

## Data production, organization and analysis

Data were produced from April 10 to July 7, 2023, through individual interviews, audio recorded in MP3 format. Previously, the researchers visited the health unit to present the study proposal to the manager and other professionals in order to request their collaboration and clarify possible doubts. Women were approached individually when they came to the unit to attend appointments and/or carry out other procedures. Thus, they were invited to go to a room provided by the manager, in which the study was briefly presented, using easy-to-understand language, inviting them to participate. With those who accepted, the interview was carried out subsequently in the same environment to ensure participant privacy and create an environment favorable to dialogue. Respecting their individuality, it should be noted that, in this environment, there were only the researcher and the participant.

We used a semi-structured script, prepared by the researchers, with two parts. In the first, nine sociodemographic and clinical variables were included, namely age, color/race, education, religion, marital status, occupation, family income, number of children and active sexual life. In the second, consisting of subjective questions to explore the object of study, we sought to understand perceptions about CC, knowledge about HPV and preventive measures. Given the characteristics of this type of instrument [32], questions were clarified and broken down by the researcher in dialogue with participants, encouraging the expression of narratives.

On average, the interviews lasted 15 minutes, and were conducted by a research team composed of nursing students enrolled in the study's proposing higher education institution (HEI). They were previously trained by nursing professors/researchers on the regular activities of two research groups at this institution, aiming to standardize procedures and avoid possible biases. Understanding that the interviews would be sufficient to understand the object of study, we decided not to use a field diary or other collection techniques.

Sociodemographic variables were tabulated in a Microsoft Office Excel®, version 2019 spreadsheet, and analyzed using descriptive statistics, highlighting the absolute values and their percentages, presented in the first paragraphs of results. Subjective data were transcribed in full to form a text *corpus*, subjected to processing and lexical analysis using *Interface de R pour les Analyzes Multidimensionnelles de Textes et de Questionnaires* (IRaMuTeQ®), version 0.7, alpha 2. Available free of charge, IRaMuTeQ® uses functionalities from the R statistical software (e.g., version 4.0.3, used in this study),

allowing statistical analyzes to be carried out on text *corpora*. This characteristic increases technical-scientific rigor, adding greater reliability to qualitative analysis [33–35].

Among the possibilities that IRaMuTeQ® offers, we used Descending Hierarchical Classification (DHC) in order to identify occurrences (forms or words) that are similar or complementary to each other. Thus, the software processed the set of interviews, demonstrating the relationships between lexical analysis and utterances contained in statements, culminating in the structuring of the *corpus* through a graphic representation (DHC dendrogram), which highlights the composition of the lexical classes and the relationships they maintain among themselves. Subsequently, the text segments (TS) of each class were carefully read to understand their meanings, considering the most representative words by the frequencies (f) of TS that presented them individually in the classes and, above all, by their chi-square ($X^2$), with $p < 0.0001$, a statistical value that symbolizes the associative strength between the words to form the classes [33–35].

To better present the subjective data, we decided to name the classes according to their content, organizing them into thematic axes. Based on these specificities, lexical analysis was guided by evidence from relevant and updated scientific literature on the topic, making it possible to interpret and discuss these data.

### Ethical aspects

We complied with Resolution 466/2012 of the Brazilian National Health Council/Ministry of Health [36], which guides and regulates ethical care in research with human beings in Brazil. Thus, institutional authorization was obtained from SESMA and approval from the *Universidade do Estado do Pará* Undergraduate Nursing Course Research Ethics Committee (REC) in January 2023, under Opinion 5.865.620 and CAAE (In Portuguese, *Certificado de Apresentação para Apreciação Ética* [Certificate of Presentation for Ethical Consideration]) 66658722.2.0000.5170. Participants' consent was obtained by reading and handwritten signing the Informed Consent Form (ICF), formally declaring acceptance. To ensure the confidentiality of their identities, alphanumeric codes were used, consisting of the letter "P" in reference to "participant", followed by the Arabic number indicating the sequence of interviews.

### Results

Among the participants, the age ranged from 23 to 64 years, with the age group from 23 to 32 years prevailing (n = 7; 35%). Regarding color/race, 14 (70%) declared themselves to be brown and, in relation to the level of education, those who reported having completed high school (n = 7; 35%) and incomplete elementary school (n = 6; 30%) predominated. There was a predominance of Catholics (n = 9; 45%) and Evangelicals (n = 6; 30%), in addition to singles (n = 8; 40%), self-employed (n = 7; 35%) and housewives (n = 6; 30%).

For monthly family income, the minimum wage in force in Brazil during 2023 (R$ 1,320.00) was used as a reference, in which 14 (70%) participants had an income lower than or equal to one minimum wage. All were mothers, and the number of children varied from one to four per woman, with those who had two children prevailing (n = 9; 45%). Regarding sexual life, 17 (85%) reported having an active sexual life (Table 1).

In the case of data generated by IRaMuTeQ®, the *corpus* consisted of 20 texts, corresponding to the set of interviews. Furthermore, 262 TS were identified, with 222 being used, making up 84.73% of the *corpus*. A total of 9,126 occurrences (forms or words) emerged, of which 1,320 were distinct words and 651 hapaxes (words whose frequency is equal to one), with an average of 34.83 occurrences per TS.

IRaMuTeQ® dimensioned and classified TS based on DHC, generating six lexical classes using a dendrogram. To present the results, the classes were analyzed in two *subcorpora*: the first *subcorpus* was composed of classes 2, 4 and 6; and the second was composed of classes 1, 3 and 5 (Fig 1). Thus, it was possible to organize three thematic axes. The first axis, consisting of classes 2, 4 and 6, brings together TS that demonstrate perceptions about CC, behaviors regarding gynecological signs and symptoms and opinions about access to information about the disease. The second axis, with

**Table 1.** Sociodemographic and clinical profile of participants (n = 20). Belém, Pará, Brazil, 2023.

| Variables | n | % |
|---|---|---|
| **Age** | | |
| 23–32 years old | 7 | 35 |
| 33–42 years old | 3 | 15 |
| 43–52 years old | 3 | 15 |
| 53–62 years old | 4 | 20 |
| ≥63 years old | 3 | 15 |
| **Color/race** | | |
| Yellow | 1 | 5 |
| White | 1 | 5 |
| Brown | 14 | 70 |
| Black | 4 | 20 |
| **Education** | | |
| Incomplete elementary school | 6 | 30 |
| Complete elementary school | 4 | 20 |
| Incomplete high school | 0 | 0 |
| Complete high school | 7 | 35 |
| Incomplete higher education | 2 | 10 |
| Complete higher education | 1 | 5 |
| **Religion** | | |
| Catholic | 9 | 45 |
| Christian (did not profess a specific Christian denomination) | 4 | 20 |
| Evangelical | 6 | 30 |
| Does not follow religion | 1 | 5 |
| **Marital status** | | |
| Married | 5 | 25 |
| Divorced | 1 | 5 |
| Single | 8 | 40 |
| Stable relationship | 5 | 25 |
| Widowed | 1 | 5 |
| **Occupation** | | |
| Retired | 1 | 5 |
| Self-employed | 7 | 35 |
| Housewife | 6 | 30 |
| General services | 2 | 10 |
| Sociologist | 1 | 5 |
| Assistant manager | 1 | 5 |
| Transport worker | 1 | 5 |
| Nursing technician | 1 | 5 |
| **Monthly family income** | | |
| ≤1 minimum wage | 14 | 70 |
| 1,5 minimum wage | 1 | 5 |
| 2 minimum wages | 3 | 15 |
| 2,5 minimum wages | 1 | 5 |
| 3 minimum wages | 1 | 5 |
| **Have children** | | |
| Yes | 20 | 100 |

*(Continued)*

**Table 1.** (Continued)

| Variables | n | % |
|---|---|---|
| No | 0 | 0 |
| **Number of children** | | |
| 1 | 3 | 15 |
| 2 | 9 | 45 |
| 3 | 7 | 35 |
| 4 | 1 | 5 |
| **Active sex life** | | |
| Yes | 17 | 85 |
| No | 3 | 15 |

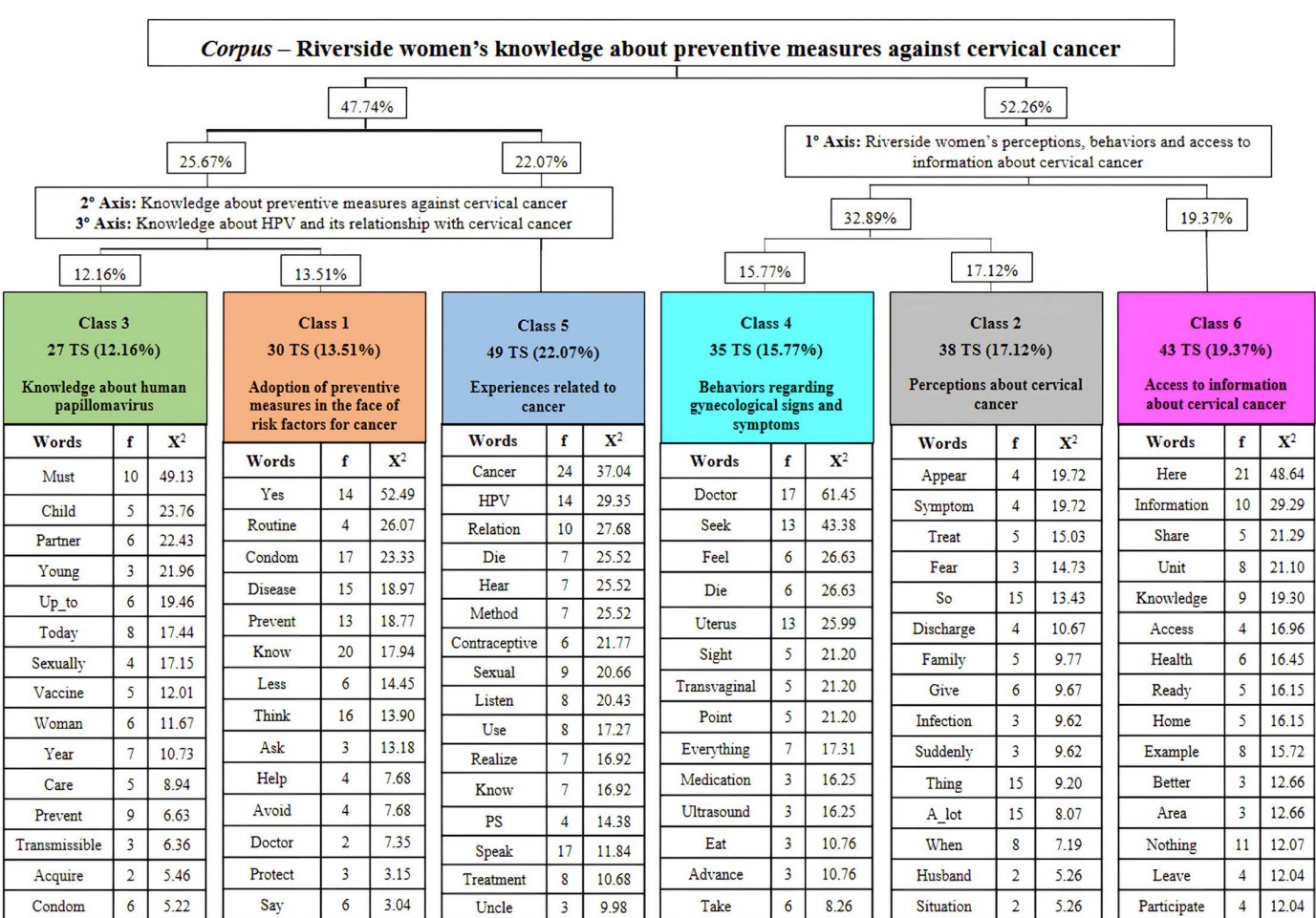

**Fig 1. Descending Hierarchical Classification dendrogram of the "Riverside women's knowledge about preventive measures against cervical cancer"** *corpus.* **Belém, Pará, Brazil, 2023.**

classes 1 and 5, demonstrates experiences related to cancer and the adoption of preventive measures in the face of risk factors for the disease. Represented by class 3, the third axis presents ideas that come from social thinking about HPV.

Based on this organization, thematic axes were named, respectively: "Riverside women's perceptions, behaviors and access to information about cervical cancer"; "Knowledge about preventive measures against cervical cancer"; and "Knowledge about the human papillomavirus and its relationship with cervical cancer" (Fig 1). The axes are presented below, highlighting the contents and main lexical characteristics of their classes as well as some emblematic excerpts.

**Thematic axis 1 – Riverside women's perceptions, behaviors and access to information about cervical cancer**

Composed of classes 2, 4 and 6, this axis demonstrates riverside women's perceptions about CC, their daily behaviors in the face of gynecological signs and symptoms that they often associate with this type of cancer as well as their opinions on access to information about CC.

With 38 TS (17.12% of the *corpus*) and 48 representative words (p < 0.0001), of which 15 (31.25%) stood out for their chi-square, class 2 builds conceptual ideas that result from women's imagination about CC. In this regard, the words "appear" (f = 4; $X^2$ = 19.72), "infection" (f = 3; $X^2$ = 9.62) and "thing" (f = 15; $X^2$ = 9.20) are closely related, as they are terms whose meanings suggest that, although women are unsure about the origin and development process of CC, they understand its occurrence as something serious and that results from the lack of treatment of changes in gynecological health over time:

> *I just know that* [CC] *kills a lot of people, it's not a good thing, it's a bad thing. It appears through inflammation, a nodule, that sort of thing. In my way of thinking, if you don't do treatment,* [if you don't carry out] *prevention, these things can cause cancer.* (P5)

> *I think it is a serious infection, which is killing a lot of people, and that we can take care of it through preventive measures* [referring to PS] *to avoid getting into this situation.* (P11)

Based on 35 TS (15.77% of the *corpus*) and 43 representative words (p < 0.0001), of which 14 (32.56%) stood out for their chi-square, class 4 presents participants' behaviors towards gynecological signs and symptoms, culminating in the decision whether or not to seek health services. The terms with the greatest strength in the class, such as "doctor" (f = 17; $X^2$ = 61.45) and "seek" (f = 13; $X^2$ = 43.38), associated with the verbs "feel" (f = 6; $X^2$ = 26.63) and "die" (f = 6; $X^2$ = 26.63), and the noun "uterus" (f = 13; $X^2$ = 25.99), are related considerably, as reports show the idea that, after the appearance of some abnormality in the body, it is necessary to seek help from a health service, especially with regard to the gynecological specialty:

> *See a doctor if you have any discomfort, as was my case,* [as] *I felt uncomfortable, a lot of cramps. I went to the doctor and he* [requested] *a transvaginal* [ultrasound]. *So, the sooner the woman seeks a doctor for assessment, to undergo tests,* [there is treatment]. *If you have a problem,* [...] *if you have had cancer in the most recent period, there is treatment.* (P14)

> *This is how I do the ultravaginal* [transvaginal ultrasound], *the ultrasound of my uterus and my breast. Because, as we have had several cases* [referring to cases in the family], *we are concerned. I'm not afraid, but I like to take care of myself.* (P20)

Consisting of 43 TS (19.37% of the *corpus*) and 56 representative words (p < 0.0001), of which 15 (26.79%) stood out for their chi-square, class 6 points to access to information about CC in women's daily lives. The words "here" (f = 21; $X^2$ = 48.64), "information" (f = 10; $X^2$ = 29.29), "share" (f = 5; $X^2$ = 21.29), "unit" (f = 8; $X^2$ = 21.10), "knowledge"

(f = 9; $X^2$ = 19.30), "access" (f = 4; $X^2$ = 16.96) and "health" (f = 6; $X^2$ = 16.45) highlight different aspects contained in reports that are consistent with participants' reduced knowledge about CC.

Access to such knowledge comes, above all, from their own experiences, from their daily relationships with people close to them and from the care they receive at the health unit, through appointments and lectures. In this context, they highlighted the fragility of access to adequate information about this pathology, characterizing this obstacle as a phenomenon that is justified by three factors: reduced frequency in sharing information in the unit; insufficiency of this information when shared; and limitations of geographic and social accessibility to the unit to participate in health education actions and spaces:

*They* [MHU professionals] *always give a lecture, but* [it is necessary] *to focus more, because I heard that there were many cases* [of the disease]. *So, even during the appointment,* [it is necessary] *to focus a little more, give better guidance to women, talk and ask if they have any questions. But there is no* [more consistent] *explanation. I came* [attended the appointment] *with a gynecologist who is very unprepared, because he doesn't give information. I even came with the nurse, because he didn't give an explanation. They should provide assistance, a better detailed explanation.* (P14)

*I believe that access to the unit is difficult, because there are some areas that, despite being an island, in the middle of the island there is water, there is a stream, you need to pass* [through] *the stream,* [through] *a river. So, I believe that access here is very difficult, and these people* [residents of the island] *may not know* [about the disease] *due to lack of information. I think it doesn't hurt to inform, talk about what you know, share your knowledge.* (P19)

### Thematic axis 2 – Knowledge about preventive measures against cervical cancer

Consisting of classes 1 and 5, this axis elucidates ideas around preventive measures against CC and experiences related to cancer, impacting the adoption of measures. Together, these classes build the idea that riverside women have preventive attitudes, which reflect individual and collective experiences on the topic over the years.

Seen in these terms, class 1 was constituted by the grouping of 30 TS (13.51% of the *corpus*), presenting 28 representative words (p < 0.0001), of which 14 (50%) stood out due to their chi-square. It is close to class 5, as both express ways of understanding the importance of barrier methods, highlighting its applicability to protect the body against infections, such as HPV infection, and thus prevent CC. Terms such as "cancer" (f = 24; $X^2$ = 37.04), "method" (f = 7; $X^2$ = 25.52), "condom" (f = 17; $X^2$ = 23.33) and "prevent" (f = 13; $X^2$ = 18.77) demonstrate that participants acquired knowledge that contributes to understanding that using condoms is a fundamental method to avoid CC.

However, although they often understand, through common sense, that it is necessary to adopt measures to prevent the emergence of diseases, they do not always know how to adopt them. Therefore, in class 1, statements also reflect the feeling of uncertainty and gaps that exist in scientific knowledge among riverside women about the correct way to protect themselves to avoid CC:

*I think that using condoms does not influence the onset of uterine cancer. Prevention is* [with] *the test we do, PS. There is another more specialized test to discover* [diagnose], *but I don't remember the name.* (P2)

[It transmits] *through sexual intercourse, these things,* [so] *you have to prevent it, do preventive care every year. I think it must be lifestyle habits, age, the person not taking care of themselves. I think condoms can help, but most of the time they don't* [help], [as] *there are those condoms that break.* (P6)

Organized by 49 TS (22.07% of the *corpus*), with 46 representative words (p < 0.0001), of which 16 (34.78%) stood out due to their chi-square, class 5 points out situations experienced by participants in relation to malignant neoplasms.

Therefore, the words "cancer" (f = 24; $X^2$ = 37.04), "HPV" (f = 14; $X^2$ = 29.35), "die" (f = 7; $X^2$ = 25.52), "hear" (f = 7; $X^2$ = 25.52), "PS" (f = 4; $X^2$ = 14.38) and "uncle" (f = 3; $X^2$ = 9.98) strengthen the perspective of transforming attitudes towards a fact that significantly marked these women's daily lives: those who know stories of people diagnosed with cancer and/or who have faced cancer treatment, especially in their families and among their acquaintances, tend to change their attitudes regarding adherence to preventive measures:

> *I take tests regularly. My sister-in-law passed away from cervical cancer when she discovered it no longer had a cure.* (P1)

> *I've met women who have lived their entire lives and never taken preventative care* [PS], *so when they go to look for* [a health service], *they already have cancer. I believe that, through prevention, you can take direction.* [But] *I never received any guidance, I never had access to this information.* (P11)

**Thematic axis 3 – Knowledge about the human papillomavirus and its relationship with cervical cancer**

This axis is represented by class 3, formed by the grouping of 27 TS (12.16% of the *corpus*), with 53 representative words (p < 0.0001), of which 15 (28.30%) stood out for their chi-square. Addressing riverside women's knowledge about HPV and its relationship with CC, this class is made up of words such as "sexually" (f = 4; $X^2$ = 17.15), "transmissible" (f = 3; $X^2$ = 6.36), "acquire" (f = 2; $X^2$ = 5.46) and "condom" (f = 6; $X^2$ = 5.22), demonstrating that they recognize HPV as the causative agent of a sexually transmitted infection (STI), which is why it is necessary to use condoms to prevent this infection, despite not having in-depth knowledge:

> *I've heard about HPV, I don't know anything very in-depth, but I think you can get it through contact during sexual intercourse. It's a lot, I forget* [information about the topic], *but I've heard about it.* (P16)

> *I've heard about HPV; I know you have to prevent it by using a condom, and the reason is having sex without prevention.* (P18)

Still in this context, it is important to highlight that, together, the words "must" (f = 10; $X^2$ = 49.13), "child" (f = 5; $X^2$ = 23.76), "vaccine" (f = 5; $X^2$ = 12.01) and "care" (f = 5; $X^2$ = 8.94) reveal that participants are aware that HPV infection can also be prevented by vaccination:

> *Highly transmissible through intimacy between men and women, which is why today there are vaccines for boys and girls from an early age, so they don't have it* [not contract the infection]. (P14)

> *I know there's a vaccine, which adolescents generally get too, and I know it's to protect themselves. In my time* [adolescence], *there was no such vaccine.* (P20)

## Discussion

The results of this study demonstrate that, although women recognized the severity of CC, they had limitations in describing its basic characteristics, demonstrating a deficient level of knowledge regarding the disease. Limitations like these were also highlighted in research with women who lived in rural communities in Malawi [37], demonstrating the lack of basic understanding about cancer in general and CC screening among participants, a reality also verified in a study with women from a traditional population in northeastern Brazil [38]. In view of this, it is important that women understand aspects related to preventive measures, as knowledge helps them to make informed decisions about screening tests and necessary care, having a positive impact on their health.

The elements related to lack of knowledge about the disease are related to several determinants, such as level of education, socioeconomic status, availability of access to health services, and women's beliefs and perceptions about concepts of health, disease and preventive measures, as pointed out in research with women from rural Indian communities of Jaipur, state of Rajasthan [13], and Javarnahalli, state of Karnataka [39]. Thus, riverside populations' low education level, their low socioeconomic level and territorial barriers they face in accessing essential services, especially health services, can have a significant impact on the way they perform self-care and reflect on their health, as evidenced by the results of this study.

A literature review on decision-making actions based on knowledge about CC identified knowledge related to signs and symptoms and preventive measures, showing reasonable knowledge of women, but restricted attitudes in seeking health services in a timely manner, despite understanding the need to carry out tests to diagnose CC early [40]. Partially differing from the findings of this review, it is important to highlight that, in the reports of this study, participants demonstrated positive attitudes that allude to the search for professional guidance and carrying out test for signs and symptoms that may make them suspect an abnormal gynecological condition, even though they did not specify the most important clinical manifestations of CC, such as pain or vaginal bleeding, highlighted in the scientific literature on the subject [40,41].

Hence, it is emphasized that the search for care in health services, without regularity and driven only by a suspected condition of abnormality or existing pathology, characterizes fragmented preventive care, with insufficient or ineffective health promotion actions, given that, in this case, the search does not occur, primarily, with the objective of preventing pathological changes in gynecological health. This context was identified in the health education experience carried out in an indigenous community with Tabajara women, in which the search for PS was found in situations of suspected bacterial or fungal genital infections, a fact that may limit the identification of lesions with carcinogenic potential by the same test [42].

Despite the positive attitudes, this highlights an important gap in the study participants' reports. This weighting is made considering that PS, with its potential to prevent and screen CC early, was rarely mentioned by women when they reported the care taken to prevent the disease, a context in which imaging tests stood out, especially transvaginal ultrasound.

Reports that indicate changes in participants' attitudes based on their experiences with people affected by CC stood out so that, with the identification of gynecological abnormalities, they are soon encouraged to seek care at a health service. From this perspective, we reflect on the repercussions of these experiences in encouraging self-care practices, emphasizing the importance of scientific knowledge so that subjects can perform them appropriately, through which they are aware of the aspects surrounding the topic, such as risk factors for illness, prevention strategies and ways to diagnose CC. This knowledge is essential as it favors women's autonomy in caring for their own bodies, avoiding conflict of information from dialogue with third parties, when this is the main source of knowledge about diseases such as CC [43].

Participants reported that, in some circumstances, they heard something about the disease, but still knew about it superficially, which is why they said they knew little about its causes and preventive measures. Other studies found a similar result, made up of a considerable number of women who had already heard about the disease and/or knew something about it, but did not know how to adequately inform a sign or symptom, preventive measures or the cause related to HPV [44,45]. Furthermore, the lack of broad and disseminated information about HPV can encourage negligent behavior, implying greater risk for the female population, since HPV infection is one of the most common STIs in the world [46].

Thus, the role of healthcare professionals in sharing information and clarifying doubts effectively is reaffirmed, taking full responsibility for promoting health and preventing illness. As strategies for this purpose, health education actions can be carried out to provide women with knowledge about the causes of CC, preventive measures and the importance of carrying out PS regularly [42,47], with the aim that such knowledge supports conscious decision-making regarding the management of their own health.

However, it is not enough to simply inform; it is necessary to pay attention to the quality of information shared. It is known that this quality is influenced by healthcare professionals' sensitivity in enabling understanding for riverside women, with educational actions in which they use accessible vocabulary on topics of individual and collective interest, valuing their geographic and sociocultural context [48,49].

Therefore, based on the statements presented here, it is understood that language adequacy to enable communication between professionals and users constitutes an aspect to be considered in the healthcare of these women. This was also highlighted by an experience report that complementarily investigate indigenous women's knowledge about CC, demonstrating that their understanding was limited and that the scenario in which they lived was not conducive to effective communication with professionals, as, in the health unit that assisted the indigenous community, this aspect was little valued in educational practices [42].

As reiterated in the literature, important limitations related to access to information about the disease include geographic and social aspects, since the main sources of information for many people, in the context of health issues, refer to what is broadcast on television media and/or appointment with healthcare professionals to whom they have access. However, not all social realities are covered by routine access to health services to obtain information about CC, due to difficulties in accessing means of maritime transport and, often, long distances between places of residence and health units, as often occurs with riverside populations, since the river dynamics and forest imposition spawn these difficulties [50].

Participants demonstrated that they understood that preventive measures are necessary in the context of the disease, but they did not know enough about which measures they should adopt, because, when asked about barrier methods, they quickly said that they would possibly not fit into this condition. Therefore, it is important to highlight that the lack of scientific knowledge about such measures means that they are not adequately prevented, a fact that can increase risk factors due to length of exposure to HPV, organism aging, sexual behaviors, among others [51].

To encourage self-care practices, this highlights the need to share, with contextualized and accessible language, that CC is a condition caused by changes in the tissue that covers the cervix due to uncontrolled proliferation of cells that deteriorate the underlying tissue. Despite having repercussions on the structure and/or functionality of adjacent organs or even more distant regions of the body, it is a slowly evolving disease that is subject to early identification, effective therapeutic intervention and adequate monitoring, resulting in a favorable prognosis and possibility of cure [13,14,52].

As vulnerable groups, in addition to being marginalized in the provision of effective public policies, it is appropriate to highlight that riverside populations are still rarely seen in health research, as evidenced by a bibliometric study that measured scientific production on these populations' health in Brazilian territory. To this end, different databases and electronic libraries were used, covering national and international literature, without choosing a time frame, study designs and languages, so as not to reduce the acquisition of materials. However, only 35 documents were included, whose first publication took place in 2007, indicating growth in production only in the four-year period from 2019 to 2022 [53].

## Study limitations

The limitations of this study refer to the impossibility, at least partial, of contextually generalizing the results to other realities nationally and internationally, since the study investigates aspects of a specific riverside population with their own characteristics, considering local geography, the territorial extension in which they live and their cultural, socioeconomic, operational and organizational peculiarities. However, it is understood that the study has the potential to encourage discussions/reflections on the topic in realities with similar characteristics in the areas of healthcare, management, teaching and research.

## Conclusions

This study analyzed riverside women's knowledge about CC and measures to prevent the disease, identifying gaps in this knowledge. Furthermore, we observed particularities of thought and attitude when adopting these measures, making

it possible to reflect on how such aspects had or could have an impact on women's daily lives in order to strengthen or weaken their health.

Considering the importance of healthcare professionals, this knowledge can give them the opportunity to rethink and redirect strategies, especially with regard to educational activities with vulnerable groups, such as riverside populations. It is understood that, in a special way, this applies to nursing professionals in Primary Health Care (PHC), given their closeness to communities and their relevant contributions to healthcare.

Therefore, in addition to supporting possible reflections on care, care management and educational activities that value the peculiar contexts of these populations, the results can encourage the proposition of new objects of study to thus develop research that makes it possible to understand or clarify aspects not covered here as well as deepen evidence about riverside women's knowledge and practices about CC.

## Supporting information

**S1 File. Appendix A – Data collection instrument.**
(PDF)

**S2 File. Renamed_86778.**
(PDF)

## Author contributions

**Conceptualization:** Ricardo Luiz Saldanha da Silva, Shirley Regina Cardoso Mendes, Bianca Silva de Brito, Bianca Pimentel de Moura, Ivaneide Leal Ataíde Rodrigues, Erlon Gabriel Rego de Andrade, Eliene do Socorro da Silva Santos, Laura Maria Vidal Nogueira.

**Data curation:** Ricardo Luiz Saldanha da Silva, Shirley Regina Cardoso Mendes, Bianca Silva de Brito, Bianca Pimentel de Moura, Ivaneide Leal Ataíde Rodrigues.

**Formal analysis:** Ricardo Luiz Saldanha da Silva, Shirley Regina Cardoso Mendes, Bianca Silva de Brito, Bianca Pimentel de Moura, Ivaneide Leal Ataíde Rodrigues, Erlon Gabriel Rego de Andrade, Eliene do Socorro da Silva Santos, Laura Maria Vidal Nogueira.

**Funding acquisition:** Ivaneide Leal Ataíde Rodrigues.

**Investigation:** Ricardo Luiz Saldanha da Silva, Shirley Regina Cardoso Mendes, Bianca Silva de Brito, Bianca Pimentel de Moura, Ivaneide Leal Ataíde Rodrigues, Erlon Gabriel Rego de Andrade, Eliene do Socorro da Silva Santos, Laura Maria Vidal Nogueira.

**Methodology:** Ricardo Luiz Saldanha da Silva, Shirley Regina Cardoso Mendes, Bianca Silva de Brito, Bianca Pimentel de Moura, Ivaneide Leal Ataíde Rodrigues, Erlon Gabriel Rego de Andrade, Eliene do Socorro da Silva Santos, Laura Maria Vidal Nogueira.

**Project administration:** Ivaneide Leal Ataíde Rodrigues.

**Resources:** Ricardo Luiz Saldanha da Silva, Ivaneide Leal Ataíde Rodrigues.

**Software:** Ricardo Luiz Saldanha da Silva.

**Supervision:** Ivaneide Leal Ataíde Rodrigues.

**Validation:** Ricardo Luiz Saldanha da Silva, Shirley Regina Cardoso Mendes, Bianca Silva de Brito, Bianca Pimentel de Moura, Ivaneide Leal Ataíde Rodrigues, Erlon Gabriel Rego de Andrade, Eliene do Socorro da Silva Santos, Laura Maria Vidal Nogueira.

**Visualization:** Ricardo Luiz Saldanha da Silva, Shirley Regina Cardoso Mendes, Bianca Silva de Brito, Bianca Pimentel de Moura, Ivaneide Leal Ataíde Rodrigues, Erlon Gabriel Rego de Andrade, Eliene do Socorro da Silva Santos, Laura Maria Vidal Nogueira.

**Writing – original draft:** Ricardo Luiz Saldanha da Silva, Shirley Regina Cardoso Mendes, Bianca Silva de Brito, Bianca Pimentel de Moura, Ivaneide Leal Ataíde Rodrigues, Erlon Gabriel Rego de Andrade, Eliene do Socorro da Silva Santos, Laura Maria Vidal Nogueira.

**Writing – review & editing:** Ricardo Luiz Saldanha da Silva, Shirley Regina Cardoso Mendes, Bianca Silva de Brito, Bianca Pimentel de Moura, Ivaneide Leal Ataíde Rodrigues, Erlon Gabriel Rego de Andrade, Eliene do Socorro da Silva Santos, Laura Maria Vidal Nogueira.

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
