## [Decision Letter · Decision Letter 0]

4 Mar 2025

PONE-D-24-13859Cervical cancer: Riverside women’s knowledge about preventive measuresPLOS ONE

Dear Dr. Saldanha da Silva,

Thank you for submitting your manuscript to PLOS ONE. After careful consideration, we feel that it has merit but does not fully meet PLOS ONE’s publication criteria as it currently stands. Therefore, we invite you to submit a revised version of the manuscript that addresses the points raised during the review process.

We look forward to receiving your revised manuscript.

Kind regards,

Fernanda Penido Matozinhos, Ph.D

Academic Editor

PLOS ONE

Journal Requirements:

“For granting two scholarships, through the Institutional Scientific Initiation Scholarship Program of the Universidade do Estado do Pará (Notice 046/2022), we would like to thank the Brazilian National Council for Scientific and Technological Development (In Portuguese, Conselho Nacional de Desenvolvimento Científico e Tecnológico) (https://www.gov.br/cnpq/pt-br), which benefited the author RLSS (Process 153917/2022-9), and Amazon Foundation to Support Studies and Research (In Portuguese, Fundação Amazônia de Amparo a Estudos e Pesquisas) (https://www.fapespa.pa.gov.br/), which benefited the author SRCM (Registration 0048397).”

**Additional Editor Comments:**

Dear Author,

After careful consideration, I feel that it does not fully meet PLOS ONE’s publication criteria as it currently stands.

The manuscript explores a very important topic, but there are major weaknesses in the results – they are not clear.

Kind regards,

Reviewers' comments:

Reviewer's Responses to Questions

**Comments to the Author**

1. Is the manuscript technically sound, and do the data support the conclusions?

Reviewer #1: Yes

2. Has the statistical analysis been performed appropriately and rigorously? 

Reviewer #1: N/A

3. Have the authors made all data underlying the findings in their manuscript fully available?

Reviewer #1: Yes

4. Is the manuscript presented in an intelligible fashion and written in standard English?

Reviewer #1: Yes

5. Review Comments to the Author

Reviewer #1: Title: Cervical cancer: Riverside women’s knowledge about preventive measures

Since this is a very specific population (riverside women from the Brazilian Amazon region), it would be interesting to highlight the nationality of this study in the title, since “Riverside” is a very generic term. Several articles in the journal PLOS ONE include the nationality of the study in the title, especially when it is a description of characteristics that are very specific to a population, without any intention of extrapolating the data, as is the case with this study.

Suggestion: “Cervical cancer: Amazonian riverside women’s knowledge about preventive measures” or “Cervical cancer: Brazilian riverside women’s knowledge about preventive measures”. Readers more familiar with the topic will know that the Brazilian Amazon region is a “hot spot” for cervical cancer, helping the reader to know whether or not this manuscript is of interest to them.

Abstract:

- Objectives, Methodology and Results – adequate

- Conclusion: Wouldn't it be better to mention some of the "gaps" in the knowledge found? Wouldn't it be better to put these gaps in the results?

Introduction:

The first two paragraphs provide information on cervical carcinogenesis, with an emphasis on the molecular role of HPV. This could be removed, as this is known information that is not of interest to this manuscript (only mentioning the role of HPV, without emphasis on molecular mechanisms). In this way, the authors could add information about studies that seek to improve screening strategies, corroborating the statistical data that the authors presented, which place the Brazilian Amazon region as a target for a closer look at this disease.

Therefore, for contextualization purposes, I suggest that quantitative studies on the prevalence of HPV infection in riverside and indigenous women, innovative methods of self-collection and molecular diagnosis of HPV for women living in isolated regions (riverside and indigenous) in Amazonas, Pará, Roraima be added, as we already have published articles on this, including in the journal PLOS ONE. These studies provide a lot of information on socioeconomic and clinical-behavioral characteristics related to HPV and other sexually transmitted pathogens that are quite comparable to the results of this study, and, taken together, provide a more complete view of the vulnerability situation of riverside women.

Methodology

- Adequate.

Results:

The descriptive sociodemographic and clinical results would be better understood in a table (first 3 paragraphs). These data are very important, as they characterize the population under study. Even though the authors have described certain aspects of the Cotijuba community in the Methodology, it is these sociodemographic data that confirm the vulnerability of these women. It is therefore necessary to make it easier for the reader to access these data so that the reader can better follow the discussion.

Discussion:

Adequate.

6. PLOS authors have the option to publish the peer review history of their article (what does this mean?). If published, this will include your full peer review and any attached files.

Reviewer #1: No

---

## [Author Response · Author response to Decision Letter 1]

16 May 2025

We thank the reviewers for their relevant contributions and for their kindness in evaluating the manuscript. When reviewing the manuscript, we took into account their suggestions/recommendations, always aiming to improve the quality of the text. In the manuscript (revised version with change control), the adjustments are highlighted with a green bar. As instructed by the journal, we would like to point out that, in addition to this version with change control, we have also attached a version without markings, the content of which is similar to that of the version with markings.

We are available for any further assistance. Below, we present the reviewers’ comments and suggestions/recommendations, with the expressions “1) ACADEMIC EDITOR’S OPINION” and “2) REVIEWER’S OPINION”, as well as our respective comments.

1) ACADEMIC EDITOR’S OPINION:

Dear author,

- After careful analysis, I feel that the manuscript does not fully meet the PLoS ONE publication criteria, as it currently stands. (AUTHORS’ COMMENT: We are aware! We have made the necessary efforts to improve the text and increase its potential to contribute to the advancement of knowledge, according to the suggestions/recommendations of the reviewers. Therefore, we understand that the text, as it stands at this time, is capable of continuing the evaluation and editorial processes, with a view to possibly obtaining acceptance for publication).

- The manuscript explores a very important topic, but there are weaknesses in the results – they are not clear. (AUTHORS’ COMMENT: Before being submitted to PLoS ONE, the text underwent rigorous language review, which was signed by the technical manager of LSB Traduções [Cadastro Nacional de Pessoa Jurídica – CNPJ: 28.340.353/0001-50], a reputable company that provides text review and translation services in Brazil. As proof of its respectability, this company was accredited by several qualified scientific journals in the field of nursing in the country, such as Acta Paulista de Enfermagem [APE – ISSN: 1982-0194], Revista Brasileira de Enfermagem [REBEn – ISSN: 1984-0446) and Revista da Escola de Enfermagem da USP [REEUSP – ISSN: 1980-220X]. Therefore, we respectfully understand that the Results meet the textual clarity required to be understood, which is why the Ad Hoc Reviewer, who evaluated this manuscript, did not make any comments or suggestions/recommendations that would indicate the lack of clarity in this section).

2) REVIEWER’S OPINION:

TITLE: Cervical cancer: Riverside women’s knowledge about preventive measures.

- Since this is a very specific population (riverside women in the Brazilian Amazon), it would be interesting to highlight the nationality of this study in the title, since “Riverside” is a very generic term. Several articles in the journal PLoS ONE include the nationality of the study in the title, especially when it is a description of very specific characteristics of a population, with no intention of extrapolating the data, as is the case with this study.

Suggestion: “Cervical cancer: Amazonian riverside women’s knowledge about preventive measures” or “Cervical cancer: Brazilian riverside women’s knowledge about preventive measures”. Readers more familiar with the topic will know that the Brazilian Amazon region is a “hot spot” for cervical cancer, helping the reader to know whether or not this manuscript is of interest to them. (AUTHORS’ COMMENT: Met! We have adjusted the title to read: “Cervical cancer: Riverside women’s knowledge in the Brazilian Amazon about preventive measures”).

ABSTRACT:

- Objectives, Methodology, and Results: Adequate. (AUTHORS’ COMMENT: Aware!).

- Conclusion: Wouldn’t it be better to mention some of the “gaps” in the knowledge found? Wouldn’t it be better to put these gaps in the results? (AUTHORS’ COMMENT: Met! We have adjusted the Results and Conclusion of the Abstract, highlighting gaps specifically in the Results).

INTRODUCTION:

- The first two paragraphs provide information on cervical carcinogenesis, with an emphasis on the molecular role of HPV. This could be removed, as it is known information that is not relevant to this manuscript (only mentioning the role of HPV, without emphasis on molecular mechanisms). In this way, the authors could add information on studies that seek to improve screening strategies, corroborating the statistical data that the authors presented, which place the Brazilian Amazon region as a target for a closer look at this disease. (AUTHORS’ COMMENT: Granted!).

- Therefore, for contextualization purposes, I suggest that quantitative studies on the prevalence of HPV infection in riverside and indigenous women, innovative methods of self-collection and molecular diagnosis of HPV for women living in isolated regions (riverside and indigenous) in Amazonas, Pará, Roraima be added, as we already have articles published on this, including in the journal PLoS ONE. These studies provide a wealth of information on socioeconomic and clinical-behavioral characteristics related to HPV and other sexually transmitted pathogens that are quite comparable to the results of this study and, together, provide a more complete view of the vulnerability of riverine women. (AUTHORS’ COMMENT: Granted! In the Introduction, in general, we have removed information on the molecular mechanisms of HIV infection and incorporated studies that address the evidence and information requested by the Reviewer. Thus, the content of this section has been expanded, aiming to better direct you to the pertinent aspects of the topic).

METHODOLOGY:

- Adequate. (AUTHORS’ COMMENT: Aware!).

RESULTS:

- The descriptive sociodemographic and clinical results would be better understood in a table (first 3 paragraphs). These data are very important, as they characterize the population under study. Although the authors described certain aspects of the Cotijuba community in the Methodology, it is these sociodemographic data that confirm the vulnerability of these women. Therefore, it is necessary to facilitate the reader’s access to these data so that they can better follow the Discussion. (AUTHORS’ COMMENT: Received! At the beginning of the Results, we created a table to detail the variables of the sociodemographic and clinical profile of the participants. Thus, in the first two paragraphs of this section, we highlight the predominant information, indicating that readers should look at the table to access the details).

DISCUSSION:

- Adequate. (AUTHORS’ COMMENT: Aware!).

---

## [Decision Letter · Decision Letter 1]

5 Apr 2026

Cervical cancer: Riverside women’s knowledge in the Brazilian Amazon about preventive measures

PONE-D-24-13859R1

Dear Dr. Saldanha da Silva,

We’re pleased to inform you that your manuscript has been judged scientifically suitable for publication and will be formally accepted for publication once it meets all outstanding technical requirements.

Kind regards,

Marianne Clemence

Staff Editor

PLOS One

Additional Editor Comments (optional):

Please could you provide a copy of the interview guide as a supporting information file with your final set of files for publication

Reviewers' comments:

Reviewer's Responses to Questions

**Comments to the Author**

1. If the authors have adequately addressed your comments raised in a previous round of review and you feel that this manuscript is now acceptable for publication, you may indicate that here to bypass the “Comments to the Author” section, enter your conflict of interest statement in the “Confidential to Editor” section, and submit your "Accept" recommendation.

Reviewer #1: All comments have been addressed

2. Is the manuscript technically sound, and do the data support the conclusions?

Reviewer #1: Yes

3. Has the statistical analysis been performed appropriately and rigorously? 

Reviewer #1: N/A

4. Have the authors made all data underlying the findings in their manuscript fully available?

Reviewer #1: Yes

5. Is the manuscript presented in an intelligible fashion and written in standard English?

Reviewer #1: Yes

6. Review Comments to the Author

Reviewer #1: I find the subject of this article very interesting. While qualitative research is not commonly featured in this journal, the topic is highly relevant, given that the Brazilian Amazon region stands out nationally in matters related to the prevalence of cervical cancer.

As a piece of qualitative research, I consider the article appropriate. The authors have made all the suggested modifications and corrections. Therefore, I deem it suitable for publication.

7. PLOS authors have the option to publish the peer review history of their article (what does this mean?). If published, this will include your full peer review and any attached files.

Reviewer #1: No

---

## [Editor Report · Acceptance letter]

PONE-D-24-13859R1

PLOS One

Dear Dr. Saldanha da Silva,

I'm pleased to inform you that your manuscript has been deemed suitable for publication in PLOS One. Congratulations! Your manuscript is now being handed over to our production team.

Kind regards,

on behalf of

Dr Marianne Clemence

Staff Editor

PLOS One